# The Impact of Reservoirs on Runoff Under Climate Change: A Case of Nierji Reservoir in China

**Baohui Men [1],\*, Huanlong Liu [1], Wei Tian [2,3], Zhijian Wu [1] and Ji Hui [4]**

[1]  Beijing Key Laboratory of Energy Safety and Clean Utilization, Renewable Energy Institute, North China Electric Power University, Beijing 102206, China; 1172211082@ncepu.edu.cn (H.L.); 1172211088@ncepu.edu.cn (Z.W.)

[2]  Key Laboratory of Water Cycle and Related Land Surface Process, Institute of Geographic Sciences and Natural Resources Research, Chinese Academy of Sciences, Beijing 100101, China; tianweiBT@163.com

[3]  University of Chinese Academy of Sciences, Beijing 100049, China

[4]  China Energy Engineering Group Guangxi Electric Power Design & Research Institute Co., Ltd., Nanning 530007, China; huiji_09@126.com

\*  Correspondence: menbh@ncepu.edu.cn; Tel.: +86-010-6177-2451

**Abstract:** Reservoirs play an important role in responding to natural disasters (such as flood and drought) by controlling the runoff. With the climate changing, the frequency and intensity of flood and drought are likely to increase. Thus, the impact of reservoirs on runoff under climate change needs to be studied to cope with future floods and drought. In this study, the Nierji Reservoir located on the Nenjiang River Basin was chosen to explore the impact. The Nenjiany River Basin is the vital water source in the water resources transfer project in Northeast China. Climate change in Nenjiang River Basin was analyzed using the 1980 to 2013 climate observations. The results show that the temperature of the basin significantly ($p < 0.05$) increased while the precipitation significantly ($p < 0.05$) decreased. Based on the result, nine kinds of different climate scenarios were set up. For different climate scenarios, the Hydroinformatic Modeling System (HIMS) model and the HIMS model with the reservoir calculation module were used to simulate runoff during the no reservoir operation period (1980–2000) and reservoir operation period (2007–2013), respectively. The impact of reservoirs on runoff under climate change is studied. Results show that the Nierji Reservoir can effectively relieve the impact of climate change on downstream runoff. When temperature increases or precipitation decreases, there are larger differences in runoff over the non-flood period, especially during periods of no reservoir operation. Reservoir operation under climate change can provide reliability in drought protection.

**Keywords:** HIMS model; reservoir operation; climate change; Nenjiang River Basin

## 1. Introduction

The fifth report (AR5) issued by the Intergovernmental Panel on Climate Change (IPCC) in 2013 stated that the global surface temperature increased by 0.85 °C from 1880 to 2012 and would increase by 0.3 to 0.7 °C in the next 30 years. In addition, the report points out that human activities are the main cause of climate warming [1,2]. Climate change can lead to changes in global hydrological processes, including melting glaciers, rising sea levels, and deterioration of the living environment [3]. The impact of climate change on water resources is mainly reflected in temperature and precipitation [4]. Precipitation is usually the most important source of water in a river, and it determines the amount of runoff [5]. Lu's research [6] showed that, on average, about 50% of the annual precipitation in China's watershed systems could be converted into runoff, slightly higher than the global average

(40%). Shi et al [7] found that precipitation changes may lead to greater changes in river runoff, especially for rivers which are not recharged by glaciers and snow meltwater rivers. According to a study by Gerten et al. [8], global average precipitation increased by 2.5% from 1901 to 2002. However, whether the global total water volume is increasing or decreasing is still a controversial issue, and the influencing factors are very complicated and vary greatly in different regions. Dai et al. [9] believed that compared with precipitation changes, the increase in evaporation caused by rising temperatures had a slower impact on runoff. Yet Trenberth et al. [10] found that increased temperatures could also enhance local precipitation by altering the thermal power of air mass and moisture transport processes. In addition, there is an important relationship between large-scale climate indices (CIs) including the El Niño-Southern Oscillation, Indian Ocean dipole, western North Pacific monsoon, and tropical cyclone activity and hydro-meteorological variables, such as precipitation, temperature, and streamflow in the tropics and extratropics. A study by Lee et al. [11] showed that the precipitation variability in Korea had a significant response to large-scale CIs. The analysis of the observed variability of all-India summer monsoon drought indices by Preethi et al. [12], for the period 1901 to 2016, revealed intensification in intensity of droughts, with an increase in percentage area of the Indian subcontinent under moderate and severe drought conditions, particularly during the post 1960 period. In short, climate has complex direct and indirect effects on river runoff.

To understand the impact of climate change on hydrological processes, the global climate models (GCMs) and the regional climate models (RCMs) are common tools for simulating climate [13]. GCMs can better simulate the average characteristics of large-scale grids, but, its outputs feature systematic biases that render them unsuitable for direct use in hydrological studies [14]. Therefore, many scholars are committed to the study of bias-correction technology. For example, Sharma et al. [15] applied the bias-correction method of gamma–gamma transformation to improve the frequency and amount of raw GCMs precipitation at the grid nodes. Wood et al. [16] used a "quantile-based" bias-correction scheme to transform the simulated and observed populations. RCMs can better simulate the characteristics of local climate information by using a finer grid within a limited domain [17]. Frei [18] used five RCMs to simulate daily precipitation statistics. Karambiri et al. [19] applied a bias correction of projected precipitation, provided by three RCMs and potential evapotranspiration projections, as input to a simple hydrological model. However, due to uncertainties in climate predictions and hydrological models, such as unknown future greenhouse gas emissions and simplification of the process, there are still errors in the predictions [20]. Therefore, some researchers investigate the impact of climate change on future water resources employing a hydrological model driven by a hypothetical climate scenario. For example, Chang et al. [21] analyzed the sensitivity of runoff to precipitation and temperature variabilities under the climate scenario of 25 hypotheses. Rehana et al. [22] simulated the water quality response of six hypothetical climate change scenarios using the water quality model QUAL2K. The results show that all hypothetical climate change scenarios would cause impairment in water quality. Walling et al. [23] estimated environmental flow containing water quality under 45 hypothetical climate scenarios. In summary, GCMs and RCMs are relatively new tools to predict climate change, but GCMs and RCMs are complex and highly uncertain. The second method is to use the hypothetical future scenario, i.e., approximate plausible variations for the future, knowing the general characteristics of the region and catchment, which is simple, quick and easy to implement. Therefore, this study uses the hypothetical future scenario to simulate climate change.

In addition to climate change, human activities are also considered to be important factors influencing runoff. Gao et al. [24] found that the contribution rate of human activities to runoff was 87.20% in the middle reaches of the Huaihe River Basin, China, while that due to climate change was 12.80%. The results of Mo et al. [25] indicated that 35.3% of the reduction in annual runoff was caused by human activities and that a 2.2% increase in annual runoff could be attributed to climate change. This finding means that human impact rather than climate exerts the dominant influence on runoff decline in the Bahe River basin, China. Numerous studies have shown that reservoirs as an important human activity have a significant impact on runoff [26–29]. A dam changes the hydrological

characteristics of the downstream river [30]. In addition, dam and water gate operations may result in the increase of runoff in the non-flood season and decrease of runoff in the flood season [31]. Moreover, the amplitude of variation in the non-flood season will be much bigger [32]. In recent decades, rising temperature and precipitation reduction in the Nenjiang River Basin in China have shown an increasingly obvious trend [33]. In December 2006, the Nierji Reservoir in the Nenjiang River Basin, a large water conservancy facility, was completed. Studying the impact of Nierji Reservoir operation on local hydrological factors under climate change can provide a theoretical basis and data support for the upstream and downstream water resources management of the Nenjiang River Basin [34].

A hydrological model is an important tool for simulating the runoff formation process to study water resources assessment, management, and utilization. Since the concept of the distributed hydrological model by Freeze and Harlan [35] was introduced, semi-distributed and distributed models have been developed and widely applied, such as the Soil and Water Assessment Tool (SWAT) model [36], the Hydrologic System Program-Fortran (HSPF) [37], and the Variable Infiltration Capacity (VIC) model [38], etc. It is important to build the hydrological model based on the hydrological characteristics of the study area. However, it is hard to customize the hydrological model according to the hydrological characteristics of the study areas with the existing models due to their fixed structures, such as the VIC model, SWAT model, HSPF, etc. [39]. The HIMS (Hydroinformatic Modeling System) model provides multiple choices describing each of the water cycling processes and accounts for the combined runoff generation mechanisms [40]. The advantage of HIMS model is its flexibility in providing alternative modules to accommodate hydrological simulation in different regions [41]. It is widely used in China, Australia, and other regions and shows good performances [40–44]. Therefore, this study applied the HIMS model to simulate runoff. When simulating runoff during reservoir operation, a reservoir calculation module needs to be established. In this research, the reservoir calculation module of HIMS model is constructed according to the actual situations of Nenjiang River Basin and Nierji Reservoir based on the reservoir calculation module establishment method of SWAT and other related models [45,46].

The main objectives of this study were to: (i) Analyze the influence of the Nierji reservoir on the downstream runoff process and set up nine climate scenarios based on climate change analysis result; (ii) apply the HIMS model to simulate the runoff in the no reservoir operation period (1980–2000), and use the HIMS model with reservoir calculation module to simulate the runoff in the reservoir operation period (2007–2013) and analyze the applicability of the model in the Nenjiang River Basin. The impact of the Nierji reservoir construction period (2001–2006) is not considered; (iii) simulate runoff during no reservoir operation period and reservoir operation period, respectively under different climate scenarios, using the HIMS model and the HIMS model with reservoir calculation module and study the impacts of climate change and reservoir operation on downstream runoff.

## 2. Study Area and Data

### 2.1. Geological Position and Hydrological Characteristic Parameters of the Reservoir

Nenjiang River Basin is located in the central and western regions of Northeast China with the geographic coordinates of 119°12′~127°54′ E and 44°02′~51°42′ N. It extends across Inner Mongolia, Heilongjiang, and Jilin provinces (autonomous regions) with a catchment area of 297 thousand km$^2$. Nierji hydro-project is located on the Nenjiang River main stream intersecting the Heilongjiang Province and Inner Mongolia Autonomous Region. Situated on the right bank of the dam-site is Nierji Town, Daur Autonomous Banner of Morin Dawa, Inner Mongolia Autonomous Region; on the left bank is Erkeqian Town, Nehe City, Heilongjiang Province with a distance of 189 km from the downstream industrial city Qiqihar. The Nierji reservoir has a controlled drainage area of 66.4 thousand km$^2$, accounting for 22.4% of the total area of the Nenjiang River Basin. The average annual runoff is 10.47 billion m$^3$, accounting for 45.7% of the Nenjiang River Basin. The geographical position is as shown in Figure 1a.

Hydrological characteristic parameters of the Nierji reservoir are as follows: The normal water level is 216.00 m; the maximum flood control operating level is 218.15 m; the flood season limited water level is 213.37 m; the dead water level is 195.00 m; the total reservoir capacity is 8.61 billion m$^3$, the flood control capacity is 2.37 billion m$^3$; the beneficial reservoir capacity is 5.97 billion m$^3$.

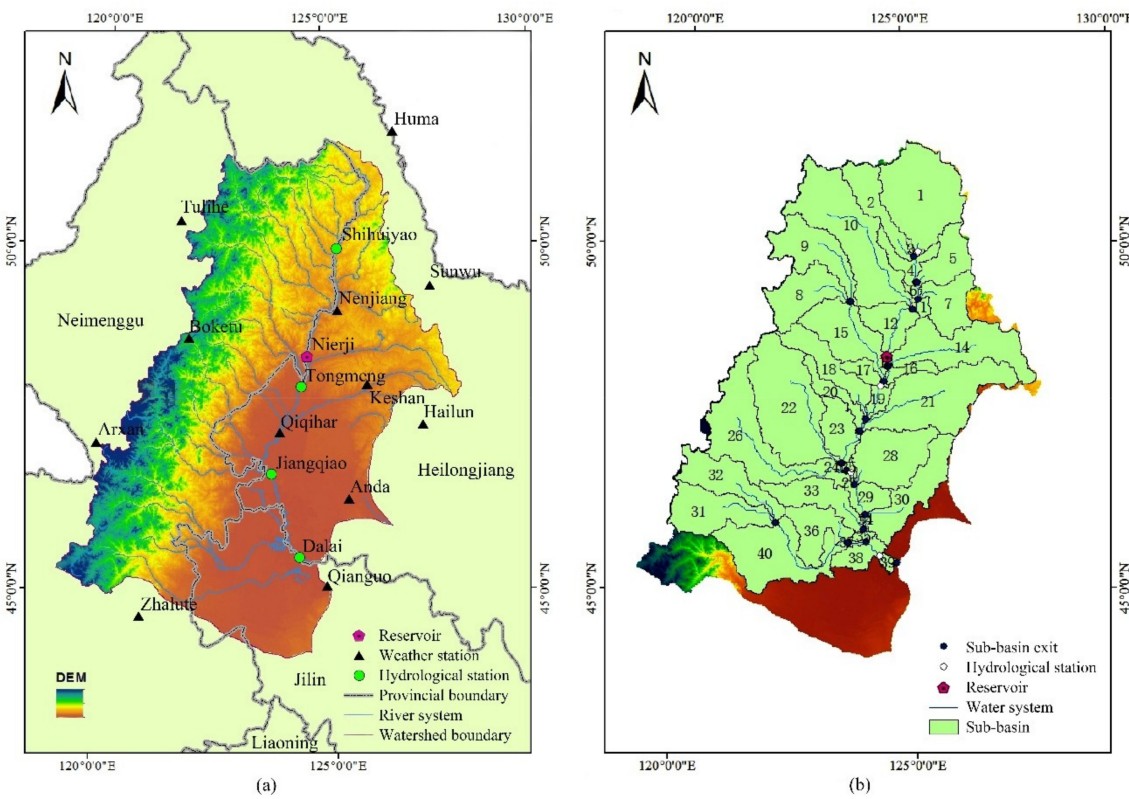

**Figure 1.** Geological position (**a**) and sub-basin delineation result (**b**) of Nenjiang River Basin.

*2.2. Data*

Construction of Nierji Reservoir divides the continuous river into several areas with different hydrologic conditions and changes the hydrologic conditions in the lower reaches of the Nenjiang River Basin. In this paper, the impact of the Nierji Reservoir operation on the lower mainstream runoff is researched. The data required for this study include the daily runoff data of the downstream hydrological stations (Tongmeng, Jiangqiao, and Dalai) before the reservoir operation (1980–2000) and after the reservoir operation (2007–2013), the DEM (Digital Elevation Model) data of the Nenjiang River Basin, and nearly 34 years of meteorological data, etc.

The runoff data are from the Hydrologic data yearbook of the People's Republic of China Hydrological Yearbook. DEM data is the modeling foundation. The DEM data of Nenjiang River Basin adopted in this paper from the Resource and Environment Data Cloud Platform (http://www.resdc.cn) are extracted from the Nenjiang River basin boundary provided by Lake-watershed Scientific Data Center, Nanjing Institute of Geography and Limnology, Chinese Academy of Science (http://lake.data.ac.cn) after being loaded to transform the coordinates to obtain the DEM data nationwide. The meteorological data includes temperature data from 12 weather stations and precipitation data from 16 rainfall stations (12 meteorological stations and 4 hydrological stations) from 1980 to 2013. The temperature data comes from the China Meteorological Data Network (http://data.cma.cn), and the hydrological station precipitation data comes from the Hydrological Yearbook of the People's Republic of China. The positions of meteorological station and hydrologic station are as shown in Figure 1a.

## 3. Methodology

### 3.1. HIMS Model

The HIMS (Hydroinformatic Modeling System) model, proposed by Liu et al., is a distributed hydrologic model [47]. The HIMS model considers a number of hydrological processes, such as precipitation, seepage, potential evaporation, runoff yield at natural storage, runoff yield in excess of infiltration, watershed flow concentration, and so on, in the simulation of runoff.

#### 3.1.1. Data Input

The initial input data of HIMS model include the sub-basin data, river network data, and meteorological data. This study uses the SWAT model to divide the sub-basins [48,49]; the result is shown in Figure 1b. According to Figure 1b, there are regions on both the left and right banks of the mainstreams in lower Nenjiang River not included in the sub-basin. Among which, the large area of swamps and wetlands on the left bank are basically inland non-contributing areas. Huolin River is on the right bank, and the lower stream of the river is the non-contributing area with a runoff mostly decreased to zero in the dry years. Hence, they are not included in the calculation of the sub-basin and basically have no influence on the model simulation result. The sub-basin data and river network data can be directly extracted from the SWAT model; the meteorological data are the data of temperature stations and precipitation stations corresponding to the sub-basins. The basic input data is then sorted according to the format requirements of the model to input data and then input them into the HIMS model to make the runoff simulation of Nenjiang River Basin.

#### 3.1.2. Constructing the Reservoir Calculation Module

There is no reservoir calculation module set in the HIMS model. It needs to be constructed by the user when performing runoff simulation during the reservoir operation period. In this research, the reservoir calculation module of HIMS model is constructed according to the actual situations of Nenjiang River Basin and Nierji Reservoir upon referring to the related documents and the reservoir calculation module of the SWAT model [50,51]. The reservoir calculation module process is shown in Figure 2. The calculation steps are as follows:

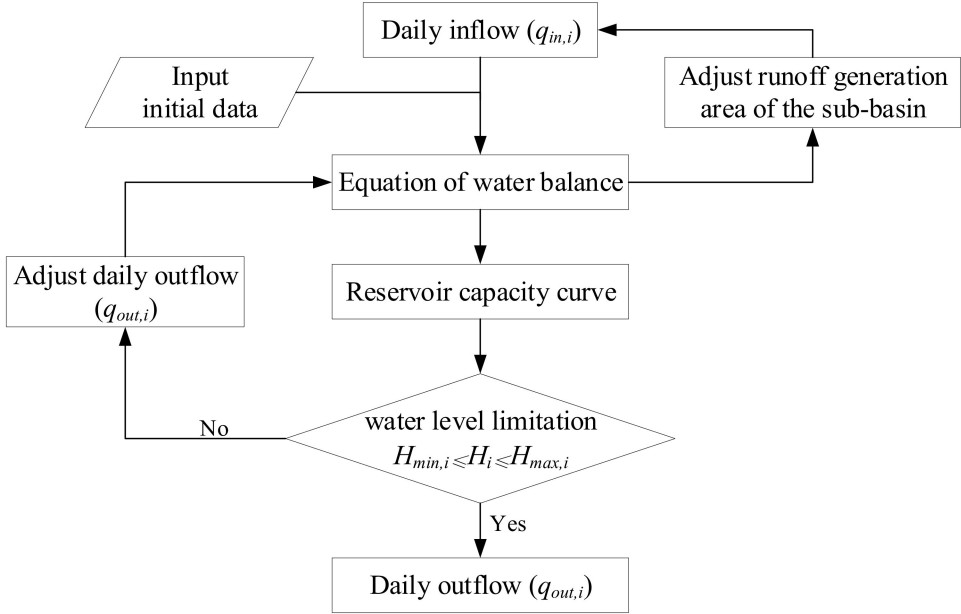

**Figure 2.** Reservoir calculation module process.

(1) Input the initial data of the reservoir, including ① serial number of the sub-basin where the reservoir is; ② initial impoundage $V_0$ of the reserior; ③ initial daily outflow $q_{out,i}$ of the reservoir, $i$ is the number of days, $i = 1, 2, \cdots, n$; ④ the effective seepage coefficient $K$ of the reservoir.

(2) The HIMS model is adopted to simulate the upstream reservoir inflow as the reservoir inflow $q_{in,i}$.

(3) Equation of water balance

$$V_i = V_{i-1} + V_{in,i} - V_{out,i} + V_{pcp,i} - V_{eva,i} - V_{seep,i} \tag{1}$$

$$V_{in,i} = q_{in,i} \times 86400 \times 10^{-9} \tag{2}$$

$$V_{out,i} = q_{out,i} \times 86400 \times 10^{-9} \tag{3}$$

$$V_{pcp,i} = P \times SA_{i-1} \times 10^{-6} \tag{4}$$

$$V_{eva,i} = 0.6E \times SA_{i-1} \times 10^{-6} \tag{5}$$

$$V_{seep,i} = 24K \times SA_{i-1} \times 10^{-6} \tag{6}$$

where $V_i$ is the reservoir impoundage of day $i$, $10^8$ m$^3$; $V_{in,i}$ is the reservoir inflow of day $i$, $10^8$ m$^3$; $V_{out,i}$ is the reservoir outflow of day $i$, $10^8$ m$^3$; $V_{pcp,i}$ is the precipitation in reservoir area of day $i$, $10^8$ m$^3$; $V_{eva,i}$ is the evaporation in reservoir area of day $i$, $10^8$ m$^3$; $V_{seep,i}$ is the seepage of the day $i$, $10^8$ m$^3$; $SA_{i-1}$ is the reservoir surface area of the day $i-1$ ($SA_0 = V_0$), km$^2$; $P$ and $E$ are, respectively, the precipitation and potential evaporation of the local sub-basin, millimeters.

(4) According to the relationship among reservoir characteristic water level, reservoir surface area, and storage capacity, the reservoir surface area–storage capacity curve and the reservoir water level–storage capacity curve can be obtained and expressed in the form of quadratic multinomial equations:

$$SA_i = a_1 V_i^2 + b_1 V_i + c_1 \tag{7}$$

$$H_i = a_2 V_i^2 + b_2 V_i + c_2 \tag{8}$$

where $SA_i$ is the reservoir surface area of day $i$, km$^2$; $H_i$ is the water level of day $i$, m; $a_1, b_1, c_1, a_2, b_2$, and $c_2$ are parameters of the reservoir surface area–storage capacity curve and the reservoir water level–storage capacity curve.

(5) If $H_i$ satisfies the restrictive condition of water level on the current day, namely $H_{min,i} \le H_i \le H_{max,i}$, $q_{out,i}$ is the daily outflow of the reservoir; otherwise, the daily outflow $q_{out,i}$ of the reservoir shall be adjusted to satisfy the conditions: ① decrease the daily outflow when $H_i < H_{min,i}$; ② increase the daily outflow when $H_i > H_{max,i}$. At the end of the dry season to the beginning of the flood season (May–June), the reservoir is operating at a low water level for flood control, and the highest water level is the flood season limited water level. In the middle of the flood season to the end of the flood season (July–September), the reservoir is operating at a high water level, and the highest water level is the maximum flood control operating level. From the end of the flood season to the middle of the dry season (From October to April the next year), the highest water level is the normal water level. The control water level of the reservoir in different periods as shown in Table 1.

**Table 1.** The control water level of Nierji reservoir in different periods.

| Month | January | February | March | April | May | June |
|---|---|---|---|---|---|---|
| Highest water level (m) | 216.00 | 216.00 | 216.00 | 216.00 | 213.37 | 213.37 |
| Minimum water level (m) | 195.00 | 195.00 | 195.00 | 195.00 | 195.00 | 195.00 |
| **Month** | **July** | **August** | **September** | **October** | **November** | **December** |
| Highest water level (m) | 218.15 | 218.15 | 218.15 | 216.00 | 216.00 | 216.00 |
| Minimum water level (m) | 195.00 | 195.00 | 195.00 | 195.00 | 195.00 | 195.00 |

(6) $q_{out,i}$ is daily reservoir outflow.

(7) Changes in the reservoir surface area influence the runoff generation area of the sub-basin where the reservoir is; which means the runoff generation area of sub-basin will increase or decrease as the reservoir surface area $SA_i$ changes. The runoff generation area of the sub-basin is calculated as:

$$S_i = S - SA_i \tag{9}$$

where $S_i$ is the runoff generation area of the sub-basin of day $i$, km$^2$; $S$ is the total area of the sub-basin where the reservoir is, kilometers squared.

### 3.1.3. Parameter Calibration

The Kling–Gupta efficiency (KGE) [52], a widely used objective function, was used to calibrate the HIMS model in this study, as shown in Formula (10). A global optimization algorithm, namely, the genetic algorithm [53], was used to find the parameter sets of the HIMS model. This algorithm is a robust and efficient search algorithm which has been widely applied to calibrate hydrological models [54,55].

$$KGE = 1 - \sqrt{(1-r)^2 + (1-\alpha)^2 + (1-\beta)^2} \quad with \quad \alpha = \sigma_s/\sigma_o \quad \beta = \mu_s/\mu_o \tag{10}$$

where $\mu_s$ and $\sigma_s$ are the mean and standard deviation of the simulations, respectively; $\mu_o$ and $\sigma_o$ are the mean and standard deviation of the observations, respectively; and $r$ is the correlation coefficient between observations and simulations. The value of *KGE* ranges from negative infinity to 1. When *KGE* is equal to 1, it indicates perfect runoff simulations.

### 3.2. Analysis of Simulation Effects

In this research, the relative error $R_e$ and Nash–Suttcliffe efficiency coefficient *Ens* are adopted to evaluate the fitting effects between the simulated results and the measured values [56,57]. Computation methods of the relative error $R_e$ and the efficiency coefficient *Ens* are, respectively, as shown in Formulas (11) and (12).

$$R_e = \frac{\overline{Q_{sa}} - \overline{Q_{ma}}}{\overline{Q_{ma}}} \times 100\% \tag{11}$$

where $\overline{Q_{sa}}$ is the average value of simulation; $Q_{ma}$ is the average value of the measured value. When $R_e < 0$, the simulation value will be smaller than the measured value; when $R_e > 0$, the simulation value will be larger than the measured value; when $R_e = 0$, the simulation value and the measured value will be perfectly matched.

$$Ens = 1 - \frac{\sum_{i=1}^{n} [Q_m(i) - Q_s(i)]^2}{\sum_{i=1}^{n} [Q_m(i) - \overline{Q_m}]^2} \tag{12}$$

where $Q_s$ is the model simulation value; $Q_m$ is the measured value of the hydrological station; n is the number of the measured value. The larger *Ens* is, the higher the model simulation accuracy will be. When $Ens = 1$, the simulation value and the measured value will be perfectly matched; when $Ens < 0$, it means the reliability of simulation value is lower than the measured value used directly; generally, when $Ens > 0.5$, the model simulation results are acceptable.

### 3.3. Analysis of Climate Change

Climate change affects runoff primarily by changing the hydrological inputs (precipitation and potential evaporation). To make the hypothetical climate scenario closer to what might happen, this study analyzed the change trends of temperature and precipitation in the Nenjiang River Basin from

1980 to 2013 before establishing the climate scenario. The Thiessen polygon is a method proposed by the Dutch climatologist A H Thiessen to calculate the average precipitation based on discretely distributed weather stations, which is widely used due to the simple calculation process [58–60]. Therefore, this study uses the Thiessen polygon method to calculate the annual average temperature and precipitation of Nenjiang River Basin. The Mann–Kendall (M-K) method can effectively confirm the mutation position of the sequence and detect the change trends [61]. Here, the M-K method is applied to detect whether the temperature and precipitation trends in the Nenjiang River Basin have changed abruptly.

## 4. Results and Analysis

### 4.1. Runoff, Climate Analysis, and Scenario Establishment

The runoff of the downstream stations before and after the Nierji Reservoir construction are shown in Table 2; the allocation proportion of the monthly runoff within the year is as shown in Figure 3. Table 2 shows that after the operation of the Nierji Reservoir, except for the non-flood period (from November to May the next year) of the Jiangqiao Station, the annual average runoff of other stations has a different degree of decline compared with that before the operation of the reservoir. Furthermore, the rate of change of runoff at each station in the flood season (June to October) is greater than that in the non-flood season. Figure 3 shows that after the operation of the Nierji Reservoir, the monthly average runoff distribution ratio (monthly runoff divided by annual runoff multiplied by 100%) in the flood season decreased, while the monthly average runoff distribution ratio in the dry season (December to March) increased. In summary, the regulation and storage capacity of the Nierji Reservoir is obvious.

**Table 2.** Statistical runoff results of stations before and after the reservoir was constructed.

| Runoff ($10^8$ m$^3$) | Period | Tongmeng | Jiangqiao | Dalai |
|---|---|---|---|---|
| Runoff amount of flood season | No reservoir operation | 15.36 | 20.22 | 19.74 |
| | Reservoir operation | 9.57 | 13.52 | 12.28 |
| | Rate of change | 37.65% | 33.15% | 37.81% |
| Runoff amount of non-flood season | No reservoir operation | 3.25 | 3.99 | 4.58 |
| | Reservoir operation | 3.24 | 4.12 | 3.75 |
| | Rate of change | 0.29% | 3.30% | 18.17% |
| Annual average runoff | No reservoir operation | 18.61 | 24.21 | 24.33 |
| | Reservoir operation | 12.82 | 17.64 | 16.03 |
| | Rate of change | 31.12% | 27.15% | 34.11% |

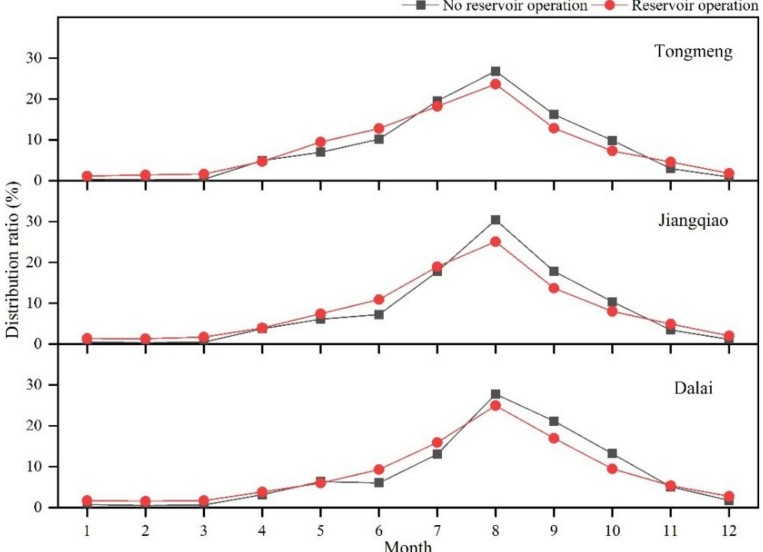

**Figure 3.** Distribution ratio of mean monthly runoff before and after the reservoir operation.

The Thiessen polygon method is used to analyze the 12 meteorological stations and 16 rainfall stations in the Nenjiang River Basin. The average temperature and precipitation in the Nenjiang River Basin from 1980 to 2013 are obtained. The trend line is added to make the analysis as shown in Figure 4. MATLAB programming is adopted to draw the M-K statistic curve of the annual average temperature and precipitation of Nenjiang River Basin; the result of which is as shown in Figure 5.

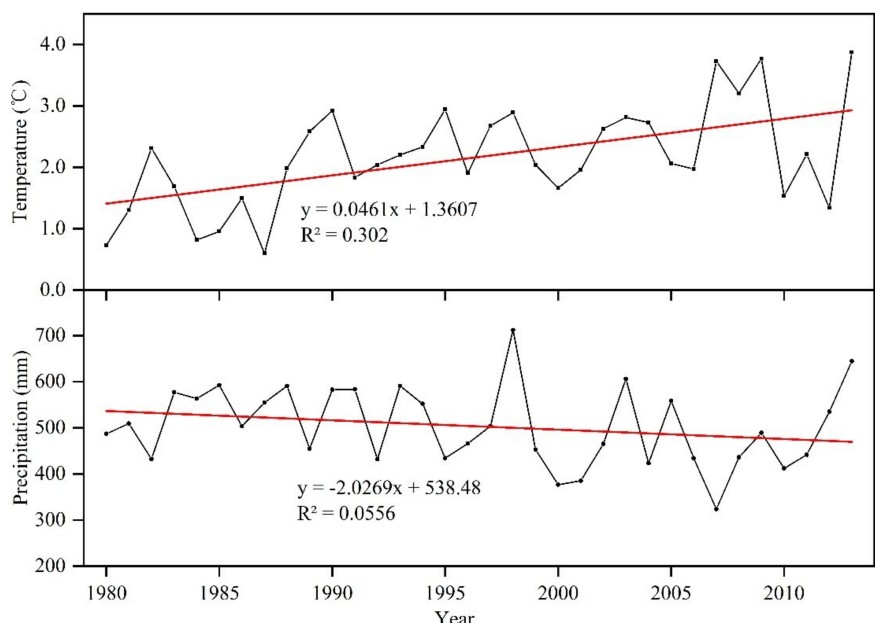

**Figure 4.** Temporal trend of the annual average temperature and precipitation.

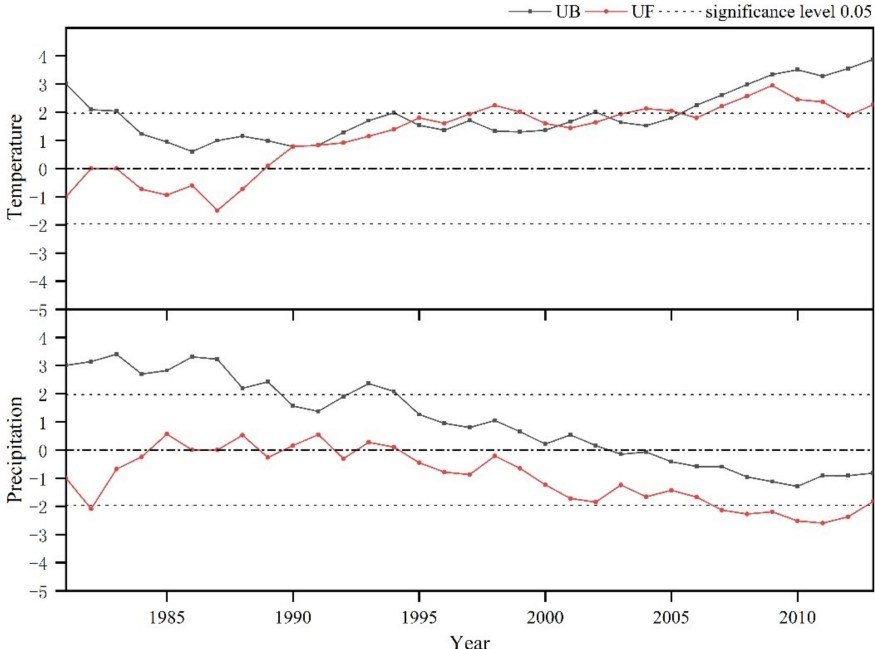

**Figure 5.** The Mann–Kendall (M-K) method analysis results of the annual average temperature and precipitation.

According to Figure 4, there is an upward trend in the temperature of Nenjiang River Basin over 34 years and a downward trend in the precipitation. According to Figure 5, statistic $UF_k$ and $UB_k$ exceed the critical line indicating that in the recent years, the upward trend of the temperature and the downward trend of the precipitation are increasingly significant. The temperature $UF_k$ curve and $UB_k$

curve intersect several times with the intersection points within the critical line interval. It means the temperature has mutations in the recent 34 years. The precipitation $UF_k$ curve and $UB_k$ curve have no intersection, which means there is no mutation in the annual precipitation of Nenjiang River Basin and it displays a steady downward trend.

According to the above analysis, the climate change trend of the Nenjiang River Basin over the recent 34 years is a temperature increase and precipitation decrease. Based on the temperature and precipitation information of the 12 meteorological stations and four hydrological stations of Nenjiang River Basin from 1980 to 2013, considering no spatial variation of precipitation intensity and spatial–temporal distribution of other climatic factors, nine kinds of different climate scenarios combined by temperature of 0 °C, 1 °C, and 2 °C increased and precipitation of 0%, 10%, and 20% decreased shall be researched as shown in Table 3. The HIMS model is adopted to make the runoff simulation and analyze the runoff change of the Nenjiang River Basin under the joint impact of the Nierji Reservoir operation and climate change.

**Table 3.** Climate scenario setting of Nenjiang River Basin.

|              | T + 0 °C | T + 1 °C | T + 2 °C |
|--------------|----------|----------|----------|
| P (1 + 0%)   | S11      | S12      | S13      |
| P (1 − 10%)  | S21      | S22      | S23      |
| P (1 − 20%)  | S31      | S32      | S33      |

### *4.2. Applicability Analysis of Model Simulation Results*

The HIMS model is adopted to simulate the daily scale runoff of Nenjiang River Basin in the no reservoir operation period (1980–2000), and HIMS model with reservoir calculation module is adopted to simulate the daily scale runoff in reservoir operation period (2007–2013). In the simulation of the runoff in the no reservoir period, 1980 and 1981 are the preheating period, 1982 to 1993 is the calibration period, 1994 to 2000 is the verification period. In the simulation of the runoff in the reservoir operation period, 2007 is the preheating period, 2008 to 2011 is the calibration period, 2012 and 2013 are the verification period. The monthly runoff simulation results in the no reservoir operation period are as shown in Figure 6; the monthly runoff simulation results in the reservoir operation period are as shown in Figure 7.

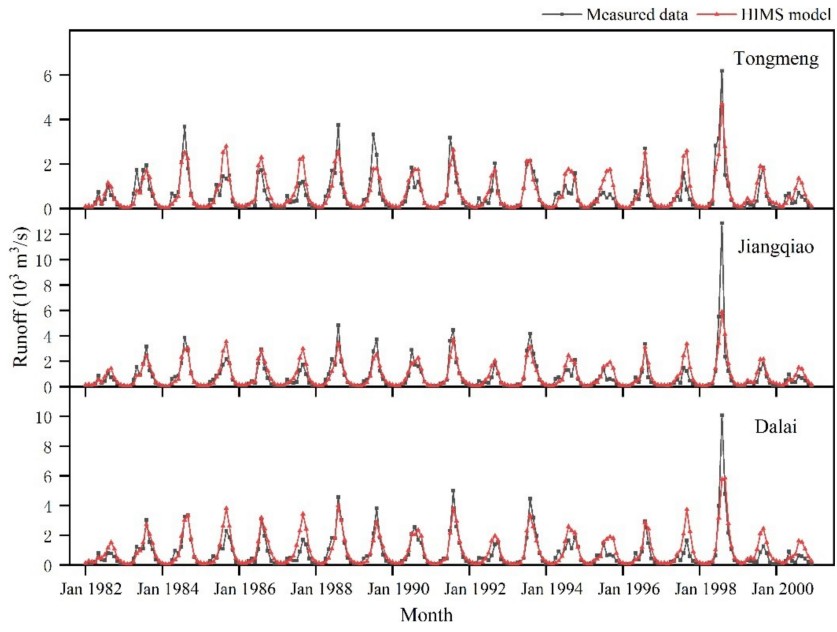

**Figure 6.** Simulation results of monthly runoff in the no reservoir operation period.

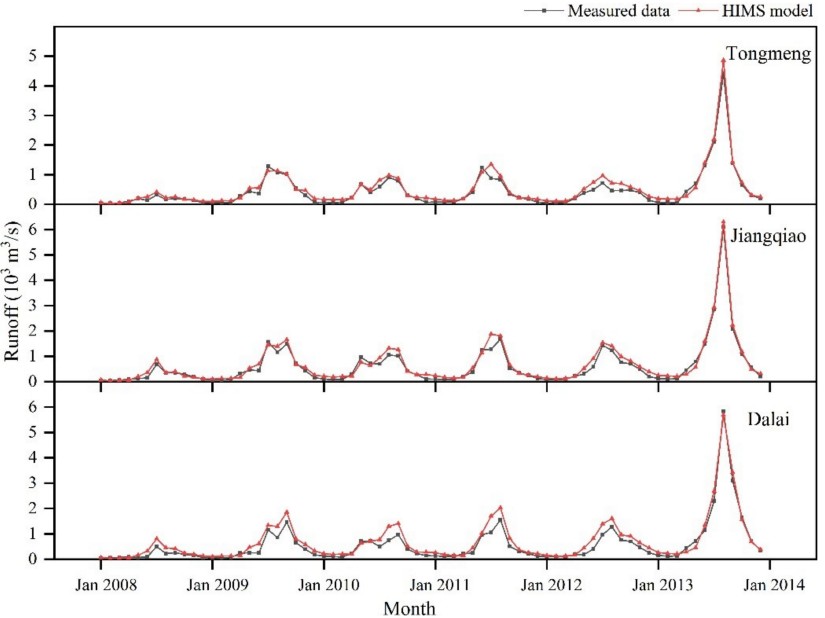

**Figure 7.** Simulation results of monthly runoff in the reservoir operation period.

The applicability of the model in the Nenjiang River Basin was analyzed. The relative error $R_e$ and Nash–Suttcliffe efficiency coefficient *Ens* were adopted to analyze the fitting effects between the simulated results and the measured values. The efficiency coefficient is computed by the mean monthly runoff; the relative error is that of mean annual runoff. Generally, the fitting result will be rather reasonable when the relative error is $|R_e| < 20\%$ and the Nash–Suttcliffe efficiency coefficient is *Ens* > 0.5. The simulated results analysis of HIMS model is as shown in Table 4.

**Table 4.** Analysis of runoff simulation results in Nenjiang River Basin.

| Reservoir | Period | Assessment | Tongmeng | Jiangqiao | Dalai |
|---|---|---|---|---|---|
| No reservoir operation | Calibration (1982–1993) | *Ens* | 0.865 | 0.897 | 0.893 |
| | | *Re*(%) | 4.00 | 3.00 | 2.70 |
| | Validation (1994–2000) | *Ens* | 0.540 | 0.612 | 0.565 |
| | | *Re*(%) | 11.28 | 4.77 | 12.56 |
| Reservoir operation | Calibration (2008–2011) | *Ens* | 0.893 | 0.885 | 0.656 |
| | | *Re*(%) | 17.50 | 14.89 | 19.56 |
| | Validation (2012–2013) | *Ens* | 0.885 | 0.887 | 0.793 |
| | | *Re*(%) | 15.77 | 11.02 | 12.69 |

According to Table 4, all analysis indexes of the runoff simulation result conform to the accuracy requirements. The reservoir calculation module design added in the HIMS model is rather reasonable and adaptable to Nenjiang River basin. It can be seen from the Nash–Suttcliffe efficiency coefficient *Ens* that the simulation effect of the validation period is worse than the calibration period in the no reservoir operation period. Although influenced by the agricultural diversion irrigation, Jianngqiao Station and Dalai Station take no impact of agricultural water into consideration when adopting the model for simulation. Therefore, the efficiency coefficient *Ens* of the calibration period in Dalai station was 0.656, which was the lowest in the reservoir operation period. Overall, it is feasible to use the HIMS model and the HIMS model with reservoir calculation module to simulate runoff in the Nenjiang River Basin during the no reservoir operation period and reservoir operation period, respectively.

*4.3. Runoff Simulation Results in Different Climatic Scenarios*

The previously hypothetical climate scenarios were introduced into the HIMS model and the HIMS model with reservoir calculation module to simulate the runoff during the reservoir operation period and the no reservoir operation period to analyze the impact of the reservoir on runoff under different climate scenarios. The results are analyzed from two aspects: mean annual change and mean month change.

4.3.1. Mean Annual Change

The mean annual runoff information of all stations under the different climate scenarios in the no reservoir operation period and reservoir operation period obtained from the HIMS model simulation are as shown in Table 5. Figure 8 shows the absolute values of the relative change rate of different climate scenarios to scenario S11 (without climate change) in the mean annual runoff simulation results of all stations.

**Table 5.** Mean annual runoff simulation results for different climate scenarios ($10^8$ m$^3$).

| Scenarios | Tongmeng | | | Jiangqiao | | | Dalai | | |
|---|---|---|---|---|---|---|---|---|---|
| | **NRO** | **RO** | **DIFF** | **NRO** | **RO** | **DIFF** | **NRO** | **RO** | **DIFF** |
| S11 | 20.77 | 16.13 | 4.64 | 25.59 | 21.62 | 3.97 | 28.53 | 22.00 | 6.53 |
| S12 | 20.01 | 16.00 | 4.01 | 24.74 | 21.36 | 3.38 | 27.63 | 21.72 | 5.91 |
| S13 | 19.39 | 15.88 | 3.51 | 24.04 | 21.07 | 2.97 | 26.88 | 21.43 | 5.45 |
| S21 | 16.71 | 14.99 | 1.72 | 20.69 | 19.26 | 1.43 | 23.10 | 19.54 | 3.56 |
| S22 | 16.30 | 14.75 | 1.55 | 20.21 | 18.91 | 1.30 | 22.57 | 19.17 | 3.40 |
| S23 | 15.94 | 14.51 | 1.43 | 19.78 | 18.58 | 1.20 | 22.11 | 18.83 | 3.28 |
| S31 | 13.55 | 12.70 | 0.85 | 16.77 | 16.01 | 0.76 | 18.71 | 16.18 | 2.53 |
| S32 | 13.31 | 12.48 | 0.83 | 16.48 | 15.72 | 0.76 | 18.39 | 15.89 | 2.50 |
| S33 | 13.10 | 12.28 | 0.82 | 16.22 | 15.43 | 0.79 | 18.10 | 15.58 | 2.52 |

Note: No Reservoir Operation (NRO); Resevoir Operation (RO); Difference between NRO and RO (DIFF).

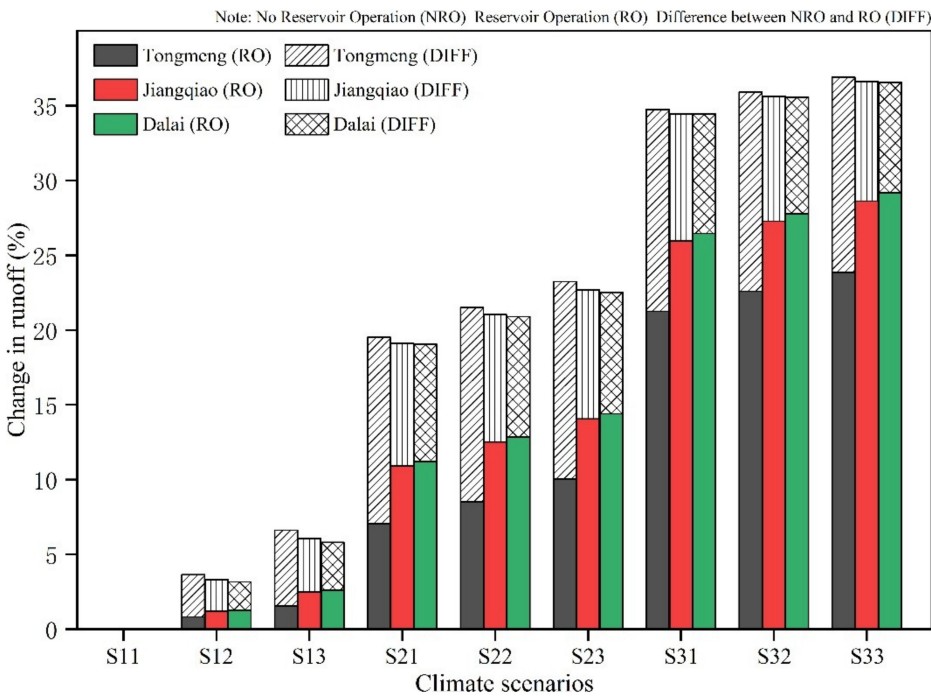

**Figure 8.** The relative change rate of mean annual runoff simulation results for different climate scenarios.

According to Table 5, without reservoir operation, the runoff of Tongmeng Station, Jiangqiao Station and Dalai Station will decrease by about 69 million, 78 million, and 83 million m³, respectively, as the temperature increases 1 °C; the runoff will decrease by about 360 million, 440 million, and 500 million m³, respectively, as the precipitation decreases 10%. With reservoir operation, the runoff will decrease by about 12 million, 27 million, and 28 million m³, respectively, as the temperature increases 1 °C and 170 million, 280 million, and 290 million m³, respectively, as the precipitation decreases 10%.

According to Figure 8, under the same climate scenarios, with no reservoir operation in each station, the relative change rates of runoff show no great difference and display a slightly decreased downward trend from upstream to the downstream. With reservoir operation, there are larger differences in the relative change rates of runoff in each station. The maximum differences in the relative change rates of the runoff at Tongmeng Station, Jiangqiao Station, and Dalai Station can be 4.76% and 5.32%; the maximum difference of Jiangqiao Station and Dalai Station is only 0.55%. Under the same climate scenarios, runoff variation of Tongmeng Station is the most obviously influenced by the Nierji Reservoir operation; the maximum difference in the relative change rate can be 13.49%. The maximum difference in the relative change rate of Jiangqiao Station and Dalai Station is rather approximate as 8.65% and 8.13%, respectively. The differences in relative change rates decrease from upstream to the downstream, which means the Nierji Reservoir operation exerts the largest impact on the nearest Tongmeng Station and a smaller impact on the remoter Jiangqiao Station and Dalai Station.

### 4.3.2. Mean Month Change

To evaluate seasonal and inter-annual change, the study compared the mean monthly runoff under the different climate scenarios in the no reservoir operation period and reservoir operation period are as shown in Figure 9. The simulated mean monthly runoff change in response to temperature change when precipitation is unchanged are shown in Figure 10 (Temperature change), and in response to precipitation change when temperature is unchanged are shown in Figure 10 (Precipitation change).

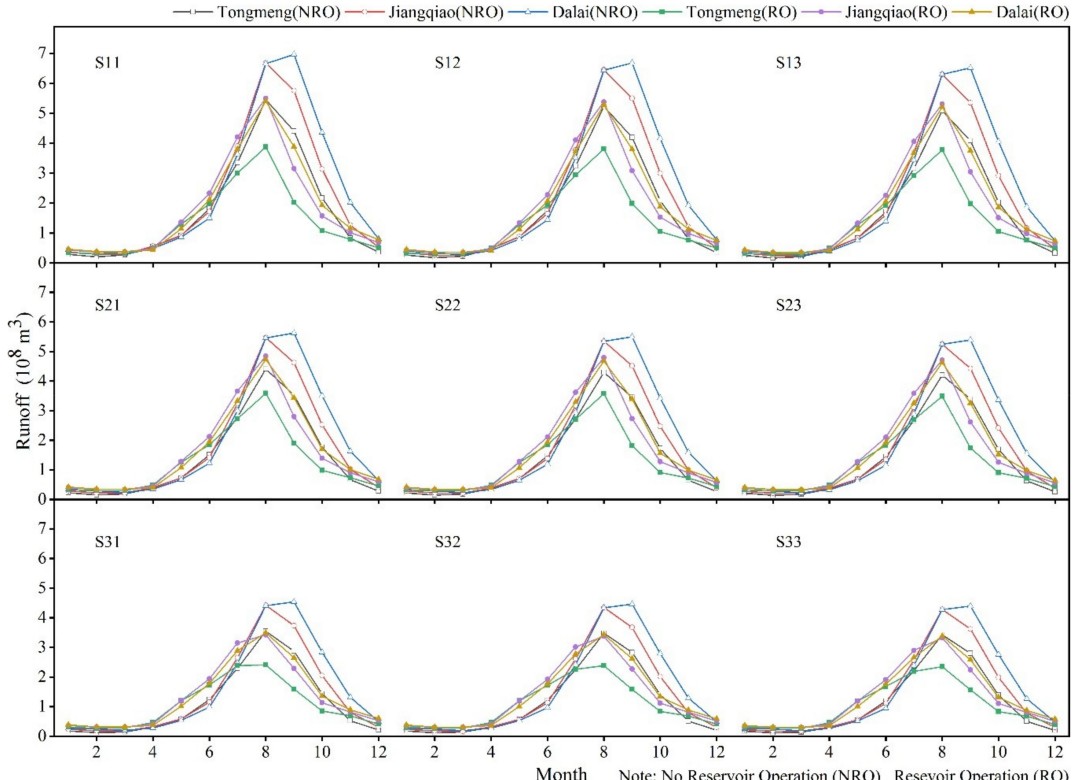

**Figure 9.** The mean monthly runoff process of nine different scenarios.

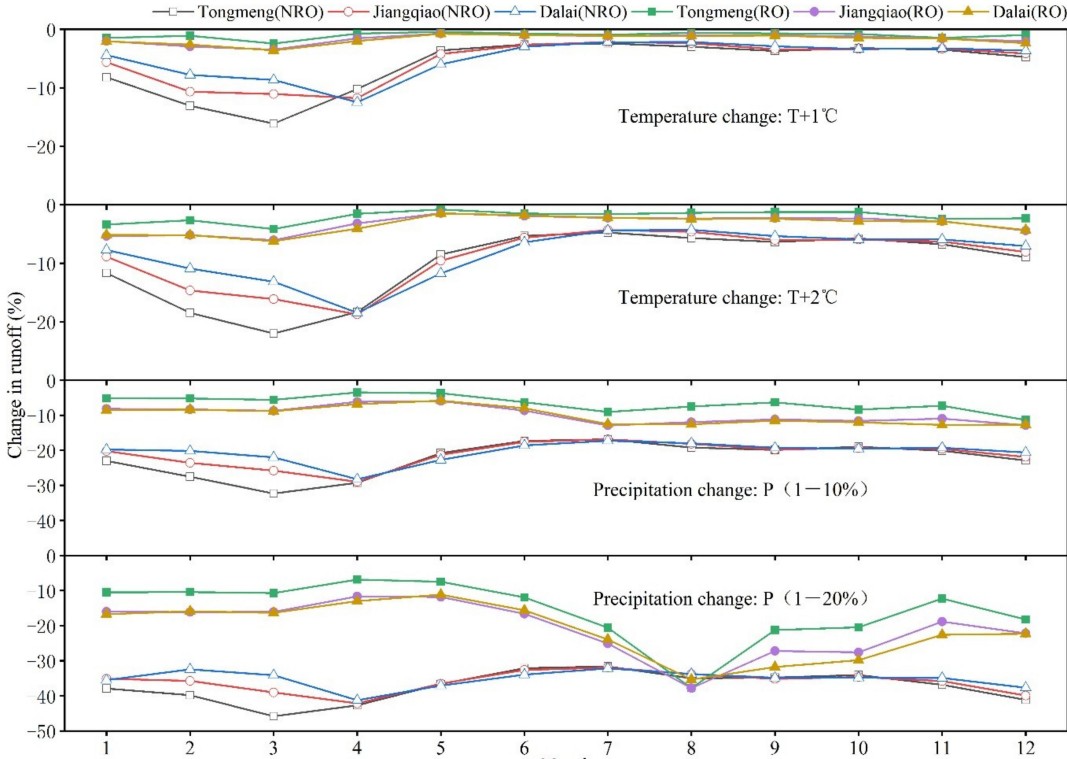

**Figure 10.** The change of mean monthly runoff under temperature or precipitation change.

Figures 9 and 10 show that:

1. Considering only the increased temperature, the runoff of all stations of Nenjiang River Basin displays a downward trend; runoff and temperature variation are negatively related. In the period of no reservoir operation, on average, when precipitation does not change, the mean monthly runoff changes by −16.12 to −2.18, −22.00 to −4.29% for temperature increases of 1, 2 °C. In the period of reservoir operation, on average, when precipitation does not change, the mean monthly runoff changes by −3.61 to −0.44, −6.23 to −0.80% for temperature increases of 1, 2 °C. When temperature increases, there are larger differences in the runoff change over the winter months.

2. Considering only the decreased precipitation, the runoff of all stations of Nenjiang River Basin displays a downward trend; runoff and precipitation are positively related. In the period of no reservoir operation, on average, when temperature does not change, the mean monthly runoff changes by −32.35 to −16.87, −45.83 to −31.61% for precipitation decreases of 10, 20%. In the period of reservoir operation, on average, when temperature does not change, the mean monthly runoff changes by −12.87 to −3.45, −37.82 to −6.83% for precipitation decreases of 10, 20%. When precipitation decreases, there are larger differences in the runoff change over the winter months. When the precipitation is decreased by 20%, there is a large change in the runoff during the flood season.

3. In the no reservoir operation period, the average runoff of all stations during the non-flood period (from November to May the next year) changes by −7.45, −11.99% for temperature increases of 1, 2 °C, while that changes by −1.81, −3.57% in the reservoir operation period. In addition, in the no reservoir operation period, the average runoff of all stations during the non-flood period changes by −23.35, −38.00% for precipitation decreased of 10%, 20%, while that changes by −7.91, −14.62% in the reservoir operation period.

## 5. Discussion

### 5.1. The Impact of Reservoirs on Runoff

Dams represent one of the most dominant forms of human impact on river systems [62] and are seen as being one of the greatest modifications to the river landscape during the Anthropocene [63]. Dam construction changes the hydrological characteristics of the basin, and the runoff process of the downstream channel will also change due to reservoir operation [64]. Zhao et al. [65] investigated the water discharge in the mainstream and seven tributaries of the Yangtze River, and found that the water discharge showed a non-significant decreasing trend at most stations except Hukou station. The results of Yang et al. [66] showed that the increase in evaporation caused by the Three Gorges Reservoir in 2003 to 2012 reduced the average downstream runoff by an average of 0.3 km$^3$ per year. This study found that after the operation of the Nierji Reservoir, the annual average runoff of downstream stations has a different degree of decline compared with that before the operation of the reservoir (Table 2). Especially at the Dalai Station, the annual average runoff decline is the most obvious, which is 34.11%. Water storage and withdrawal can directly reduce water discharge. Furthermore, reservoir construction increases the surface area of the water, thereby promoting evaporation in the basin. These reasons all lead to a reduction in runoff downstream of the reservoir. Similar results were also found in Dongjiang River [67], Kuye River [68], and Ebinur Lake [69]. These implied that the reservoir had an important influence on downstream runoff.

On the other hand, the reservoir has affected downstream runoff especially significantly through seasonal regulation of its water storage [70]. Chen et al.'s research [71] suggested that it is possible that storage delays in the dams affect the monthly distribution of discharge. Yang et al. [72] found that reservoirs usually impound water during the latter half of the wet season (decreasing trend in discharge from August to November) and release water during the driest months (increasing trend in discharge in January and February) within the Yangtze Basin. In addition, Song et al. [73] found that the reservoir operation also changes the flood regime of the Sanchahe River Basin, reducing the flood risk in terms of both in flood peak flow and volume. Similarly, the results of this study show that the Nierji Reservoir operation can make the annual runoff process of the Nenjiang River Basin a gentle trend (Figure 3). The monthly average runoff distribution ratio in the flood season (June to October) decreased, while the monthly average runoff distribution ratio in the dry season (December to March) increased. The rate of change of runoff at each station in the flood season was greater than the rate of change in non-flood season (Table 2). The large difference in the percentage change in the flood season indicates that the regulation and storage capacity of the Nierji Reservoir is obvious.

### 5.2. The Impact of Reservoirs on Runoff Under Climate Change

With climate change, the frequency and intensity of floods and droughts may increase. It may cause changes in the impact of dams and reservoirs on runoff [74]. In general, climate change affects runoff by changing the hydrological inputs (precipitation and potential evaporation) [75]. Li et al. [76] found that temperatures across the Nenjiang River Basin had steadily and significantly increased from 1960 to 2009, while precipitation had declined from 1990 to 2009. Moreover, Feng et al.'s results [33] indicated that the Nenjiang River Basin showed an obvious increasing trend of temperature increase and precipitation decrease in the recent decades. Similarly, the results of Figures 4 and 5 of this study verify this conclusion. In addition to this conclusion, the study also found that the mutation points of temperature change were mainly concentrated in the late 1990s while the precipitation change has no mutations. These changes are mainly caused by human activities (e.g., river water withdrawals for irrigation, land use changes) [77].

What effect does the reservoir have on runoff under climate change? Li et al. [78] attempted to investigate potential impacts of future climate change on streamflow and reservoir operation performance in a Northern American Prairie watershed. The results show that the current reservoir operating rules under climate change can provide a high reliability in drought protection and flood

control. Using a case study for California's Central Valley Project and State Water Project systems, Brekke et al. [79] found that climate change would influence flood control constraints on water supply operations. This paper is devoted to studying the impact of reservoirs on runoff under climate change. The results of this study show that the relative change rates of the runoff without reservoir operation in all stations of Nenjiang River Basin under climate change are higher than the relative change rates of the runoff with reservoir operation (Figure 8). Without reservoir operation, the annual runoff is influenced by climate change; and the change is rather intense. The maximum relative change rate is 36.90% without reservoir operation and 29.17% with reservoir operation. The larger difference indicates that the Nierji Reservoir operation can relieve the impact of climate change on the downstream runoff of the reservoir. Chang et al. [21] found that runoff change was more sensitive to precipitation change than to temperature change. Similarly, Figure 10 of the study can also prove this point. In addition, the results of Figure 10 indicate that when temperature increases or precipitation decreases, there are larger differences in runoff change over the non-flood period, especially during periods of no reservoir operation. This finding indicates that the downstream of the Nenjiang River Basin will be drier in the dry season under climate change during the no reservoir operation period. When the precipitation is decreased by 20%, there is a large change in the downstream runoff in the flood season during periods of reservoir operation (Figure 10). This is mainly because the Nierji Reservoir may be in a low water level operation state, and the reservoir needs to store more water to supply the water-use sectors during the dry season.

### 5.3. Reasons for the Impact of Dams and Reservoirs on Runoff

In 2007, the concept of connectivity had been further elaborated in hydrology by Bracken, where connectivity is seen to act on different spatial directions, i.e., longitudinal (river channel), lateral (hillslope/floodplain-channel), and vertical (surface–subsurface) connectivity [80]. Under natural conditions, connectivity of the hydrological systems is driven by geological and geomorphological conditions, climate and biota [81,82]. In addition, human activities, such as dam development and reservoir operation, may have an impact on structure and function, and, hence, connectivity, of geomorphic systems [83]. The dam changed the law of river continuity and destroyed the connectivity of the river. The flow velocity, depth of water and flow boundary conditions of the downstream river have changed due to the dam construction [62]. After the dam is built, the reservoir will be dispatched according to the needs of flood control, water supply, and power generation. It generally leads to an increase in runoff during the dry season of the downstream river and a decrease in runoff during the wet season [84]. In addition, the study also found that Nierji Reservoir operation exerts the largest impact on the nearest Tongmeng Station and has less impact on the remoter Jiangqiao Station and Dalai Station (Figure 8). Dis-connections are more present in the hydro–geomorphological system than connections due to dams in the river [80]. This (dis)-connectivity may be the reason why runoff yield is not just the sum of sources [85]. Wainwright et al. [86] found that connectivity was strongly influenced by climatic seasonality. As this study shows, when only considering temperature increase or precipitation decrease (Figure 10), there are larger differences in runoff change over the winter months [87].

On the other hand, a great issue in the operation of reservoir is sedimentation [88]. Reservoir operation is affected by tributaries' entrance sandbars, and the discharge building is affected by sedimentation before the dam [89]. Therefore, sedimentation has a non-negligible effect on the operation of the reservoir. This study simulates the influence of the Nierji Reservoir in the Nenjiang River Basin on runoff under climate change without considering sedimentation. This is the inadequacy of this study. Precipitation is the driving force of most water erosion processes, through detachment of soil particles and creation of surface runoff [90]. Angulo-Martínez et al.'s research [91] indicated that estimating rainfall erosivity is the key to assessing soil erosion risk. However, Zhong et al. [92] collected the distribution of rainfall erosivity in the Nenjiang River Basin, the second Songhua River Basin, and the middle reaches of the Songhua River Basin. The results showed that the rainfall erosivity

is the lowest in the Nenjiang River Basin. In addition, according to Nie's survey [93], the cities included in the Nenjiang River Basin have carried out two projects of returning farmland to forests in 2000 and 2014, respectively, which will effectively reduce soil loss in the basin. Therefore, sedimentation is not considered by the study but will definitely affect the results, but the impact may not be very large.

## 6. Conclusions

After the operation of the Nierji Reservoir, the annual average runoff of downstream stations has a different degree of decline compared with that before the operation of the reservoir. The large difference in the flood season indicates that the regulation and storage capacity of the Nierji Reservoir is obvious. Through climate analysis, in recent decades, Nenjiang River Basin displays an increasingly obvious trend of temperature increase and precipitation decrease. The relative change rates of the runoff without reservoir operation in all stations of Nenjiang River Basin under climate change are higher than the relative change rates of the runoff with reservoir operation. The difference indicates that the Nierji Reservoir operation can relieve the impact of climate change on the downstream runoff of the reservoir. When temperature increases or precipitation decreases, there are larger differences in runoff change over the non-flood period, especially during periods of no reservoir operation. The reservoir operation under climate change can provide reliability in drought protection. Finally, the impact of the reservoir should be considered in the simulation of runoff. Therefore, in the future water resources management forecast, the model considering the reservoir impact should be used for simulation.

**Author Contributions:** B.M. conceived the research theme; H.L. provided data and designed the analytical approach proposed; W.T. and Z.W. performed analysis; H.L. and J.H. wrote the paper.

**Funding:** This work was supported by the National Key R&D Program of China (Grant No. 2016YFC0401406), and the Famous Teachers Cultivation Planning for Teaching of North China Electric Power University (the Fourth Period).

**Conflicts of Interest:** The authors declare no conflict of interest.

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
