# Peer review of "The Impact of Reservoirs on Runoff Under Climate Change: A Case of Nierji Reservoir in China"

_water, doi:10.3390/w11051005_

Round 1

Reviewer 1 Report

The manuscript has been revised well in according to my comments. Therefore, in my opinion, the paper can be accepted in this form. 

Author Response

Dear Reviewer:

We appreciated very much to Reviewers for your positive comments, the reviewers’ comments and suggestions are very important to improve the manuscript, and the authors thank the reviewers a lot. 

Reviewer 2 Report

At the indroduction the frame of climate change must be improved. Author have to make an extensive literature review about climate change and most recent climate tools  such as Regional Climate Models (RCMs), eg. Cordex. Also compare the scenario they use with these models (advantages-limitation ect.). Also clearly present the novelty of this study.

A great issue in the operation of reservoir is also sedimentation. In the text there is no reference to this issue. Also many papers regards the effect of climate change of soil loss and sedimentation so you have to add some comment and estimate if these effectwill influence your study.

Finally, at the discussion a more extensie comparison with similar studies and results must be done

Author Response

Dear Reviewer:

We appreciated very much to Reviewers for your positive comments, the reviewers’ comments and suggestions are very important to improve the manuscript, and the authors thank the reviewers a lot. The relative corrections were listed below.

1. At the introduction the frame of climate change must be improved. Author have to make an extensive literature review about climate change and most recent climate tools such as Regional Climate Models (RCMs), eg. Cordex. Also compare the scenario they use with these models (advantages-limitation ect.). Also clearly present the novelty of this study.

The introduction of this paper has been extensively revised in lines 39-106 in the revised manuscript.

2. A great issue in the operation of reservoir is also sedimentation. In the text there is no reference to this issue. Also many papers regards the effect of climate change of soil loss and sedimentation so you have to add some comment and estimate if these effect will influence your study.

It has been added in lines 487-501 in the revised manuscript.

3. At the discussion a more extensive comparison with similar studies and results must be done.

The discussion of this paper has been extensively revised in lines 406-466 in the revised manuscript. 

Reviewer 3 Report

The authors performed hydrological modeling and comparative analyses for reservoir water variability for the Nierji Reservoir located on Nenjiang River Basin, China, using the Hydroinformatic Modeling System (HIMS) model. The authors demonstrated that the proposed reservoir showed the impact of climate change on the downstream runoff variation with respect to precipitation and temperature. In my opinion, these aspects are not substantial enough or novel enough to warrant publishing the manuscript in its present form. Although I realize that it takes a great deal of time and effort to prepare a paper for submission, I have some concerns about the data and methodology. Too much of the analysis and conclusions are similar to established literature on the simulation techniques for hydrologic variability, including the reference papers except for the conjunctive use of reservoir calculation. Also, the current state presents mostly a data analysis without deeper investigation and potential implications of the results studied in the paper. There is no novel methodological approach or an unprecedented finding in the current author’s manuscripts. In a revised manuscript, the authors must further develop the novel aspects of their analysis and findings. My further comments below contain some suggestions for extensions to the current manuscript.

Author Response

Dear Reviewer:

We appreciated very much to Reviewers for your positive comments, the reviewers’ comments and suggestions are very important to improve the manuscript, and the authors thank the reviewers a lot. The relative corrections were listed below.

Major revisions recommended:

1. It is one of the most difficult works to develop a hydrologic variable simulation model since the hydrologic variation has a tendency to fluctuate depending on its condition of gauging stations, anthropogenic impacts, observational accuracy, observational devices, and so on. Therefore, the authors should describe the accuracy of the reservoir water level data in terms of the above conditions in order to be a much more affirmative contribution to the literature.

Thank you very much for your comments. This opinion is helpful for improving the level of the article. The reservoir calculation module process is shown in the figure. Related revisions on page 5, line 196-204.

2. The most important prerequisite for developing a good hydrologic variable simulation model is to obtain an unimpaired observational time series. First of all, I think the data is needed to be more clarified in terms of maintaining natural data time series. According to a global reservoir data, it is found that the records used in this study could be regulated. This could be a critical issue on developing reservoir water simulation model, since the hydrologic modeling of the time series are solely based on the observational data, and if some reservoir water levels are regulated in the same (or similar) rules, they can dominate the corresponding simulation model. Consequently, if the regulated reservoir water level dominated the proposed hydrologic model, the developed model could not be regarded as a natural hydrological variability model. This concern is also needed to be resolved with the spatial homogeneity of observational devices and stations. I understand it is difficult to obtain spatially homogeneous station records. However, this point should be addressed or discussed because this paper is aimed to develop a robust and useful hydrologic tool for simulating reservoir water variability.

The HIMS model is applied to simulate the runoff in the no reservoir operation period (1980-2000), and the HIMS model with reservoir calculation module is used to simulate the runoff in the reservoir operation period (2007-2013). Related revisions on page 1, line 24-26; page 3, line 109-112; page 3, line 119-126; page 11, line 374-377; page 13, line 404-408; page 13, line 410-414;

Hydrological characteristic parameters of the Nierji reservoir are as follows: the normal water level is 216.00m; the maximum flood control operating level is 218.15m; the flood season limited water level is 213.37m; the dead water level is 195.00m; the total reservoir capacity is 86.1 hundred million m³the flood control capacity is 23.68 hundred million m³; the beneficial reservoir capacity is 59.68 hundred million m³. Related revisions on page 3, line 145-149.

At the end of the dry season to the beginning of the flood season (May-June), the reservoir is operating at a low water level for flood control, and the highest water level is the flood season limited water level. In the middle of the flood season to the end of the flood season (July-September), the reservoir is operating at a high water level, and the highest water level is the maximum flood control operating level. From the end of the flood season to the middle of the dry season (From October to April the next year), the highest water level is the normal water level. The control water level of the reservoir in different periods as shown in Table 1. Related revisions on page 6, line 239-246.

3. The authors used correlation coefficients between the measured and simulated hydrologic variable time series. However, it is necessary to describe the calculation method for correlation coefficient since the widely-used Pearson’s approach should be checked weather the dataset exhibit normal distribution or not. In this case, Spearman’s rank test is commonly used as an alternative calculation method. Additionally, the authors should address the confidence level for the correlation analysis and correct the correlation coefficient to the coefficient of determination.

Thank you very much for your comments. Your opinion is correct. The relative error                                                and Nash-Suttcliffe efficiency coefficient  can evaluate the fitting effects between the simulated results and the measured values. So, we consider using relative error  and Nash-Suttcliffe efficiency coefficient . Related revisions on page 7, line 269-271; page 12, line 388-399.

4. The conclusions made are sound. However, I would like to see a little more discussion on how the proposed hydrologic modeling can be used for water managements. Particularly, what further analyses and processes are required to utilize the proposed model to regional or local water management? Although the authors have described following limitations and highlighted the objective of this paper is to improve understanding of reservoir water variation, more elaborated implications of this study might be helpful to readers of this journal.

Page 16, Line 471-476 were added to analyze the results; Page 17, Line 536-542 were added in discussion. Conclusion and Abstract were extensively revised. Related revisions on Page 1, Line 30-36; Page 19, Line 581-606.

5. Although it is difficult to address the physical mechanism of the climatic linkage between the largescale climate fluctuation and the regional hydrologic variability, the authors should describe what are the principle drivers of the impacts of the climate variability on the hydrologic parameters of the proposed reservoir as well as identify what are the specific definitions of the climate change.

HIMS model is a distributed hydrologic model. The model considers a lot of hydrological processes such as precipitation, seepage, potential evaporation, runoff yield at natural storage, runoff yield in excess of infiltration, watershed flow concentration and so on in the simulation of runoff. Climate change affects runoff primarily by changing the hydrological inputs (precipitation and potential evaporation). Related revisions on Page 5, Line 176-181.

Minor revisions recommended:

1.but, it outputs feature but, its outputs feature

It has been modified on Page 2, line 60.

2. Specify the “The second approach”, “This method”, and “the second method”.

It has been modified on Page 2, line 79-83.

3. Rephrase the paragraph for better understanding.

It has been modified on Page 3, line 116-132.

4. The descriptions of the equation and explanations are need to be rephrased concisely in order for readers to understand.

It has been modified on Page 5, line 205-256.

5.correlation coefficient coefficient of determination

It has been modified on Page 7, line 269-271.

6. Give the reason why Thiessen polygon was applied in this present analysis despite the fact that this approach could not account for the elevation effects of the observational locations as opposed to the isohyetal method.

It has been modified on Page 8, line 286-298.

7. Remove the paragraphs of the general description of the existing methods for simplification.

It has been deleted on Page 8, line 299-326.

8. Explain “runoff distribution ratio”

It has been modified on Page 9, line 336-337.

9. Figure 3: “Change trend” → “Temporal trend”

It has been modified on Page 10, line 352.

10 “no mutations. And these change” → “no mutations. These change”

It has been modified on Page 17, line 519.

Round 2

Reviewer 2 Report

The manuscript in the revised Form can be accepted for publication in Water Journal

Author Response

Dear Reviewer:

We appreciated very much to Reviewers for your positive comments, the reviewers’ comments and suggestions are very important to improve the manuscript, and the authors thank the reviewers a lot.

    English language and style have been extensively revised in the revised manuscript. 

Reviewer 3 Report

Attached is the review comment.

Author Response

Dear Reviewer:

We appreciated very much to Reviewers for your positive comments, the reviewers’ comments and suggestions are very important to improve the manuscript, and the authors thank the reviewers a lot. The relative corrections were listed below.

1.       Related to above subject, I recommend the following recent literature in their introduction.

It has been added on Page 2, line 52-60.

This manuscript is a resubmission of an earlier submission. The following is a list of the peer review reports and author responses from that submission.

Round 1

Reviewer 1 Report

The authors show in this paper an interesting methodology to analyse the role of reservoirs to control the runoff under a climate change scenario. It is applied to an important infrastructure located in China: the Nierji Reservoir. Also a new model to simulate the reservoir operation was developed.

The manuscript is well-presented but, in my opinion, some issues are not clear to me to accept this work in its current form. These issues are:

·         Extensive editing of English language and style is required:

o   line 108: “compotation”;

o   line 207: “Tyson”;

o  

·         Line 98: the reference is not well defined;

·         The calibration stage of the model (paragraph 3.1.3.) is not clear. In my opinion more details concerning the calibration are needed: methods, objective functions...

·         The equations 10 and 11 are not clear. In Eq. 10: does Qs and Qm refer to the entire time series or to an isolated values (peak values, …)? In Eq. 11: the authors define the parameter i but they don’t use it in the expression;

·         Table 1 is not well defined: flood season, non-flood season and annual average are not Runoff. They are different scenarios.

Author Response

Dear Review:

We appreciated very much to Reviewers for your positive comments, the reviewers’ comments and suggestions are very important to improve the manuscript, and the authors thank the reviewers a lot. The relative corrections were listed below.

1. Extensive editing of English language and style is required:

o   line 108: “compotation”;

o   line 207: “Tyson”;

o   …

It has been modified in line 111, 240 in the revised manuscript. “Compotation” was rewritten as “calculation”; “Tyson” was rewritten as “Thiessen”. In addition to what you have mentioned, other places have been modified, too.

2. Line 98: the reference is not well defined;

It has been modified in line 102 in the revised manuscript.

3. The calibration stage of the model (paragraph 3.1.3.) is not clear. In my opinion more details concerning the calibration are needed: methods, objective functions...

It has been added in lines 165-174 in the revised manuscript

4. The equations 10 and 11 are not clear. In Eq. 10: does Qs and Qm refer to the entire time series or to an isolated values (peak values, …)? In Eq. 11: the authors define the parameter i but they don’t use it in the expression;

It has been explained in lines 181-188 in the revised manuscript.

5. Table 1 is not well defined: flood season, non-flood season and annual average are not Runoff. They are different scenarios.

It has been explained in line 237 in the revised manuscript 

Reviewer 2 Report

The paper describes the impact of reservoirs on runoff under climate change. In particular, the manuscript presents an interesting research aimed to study the climate change in Nenjiang River Basin (Northeast China) were analyzed using the 1980-2013 climate observation.

In my opinion, the paper describes exhaustively the main goals of research. It is interesting enough to warrant publication in Water journal. However, below are a few suggestions for minor revisions to the manuscript.

ü  In the introduction section, the authors are encouraged to revisit the literature review and to expand on some recent key studies around the "climate change" thematic, improving the background. Moreover I suggest you to mention the importance of studying the impact of climate change to local scale. The following studies could be a good source of references:

·         Elferchichi, A., Giorgio, G. A., Lamaddalena, N., Ragosta, M., & Telesca, V. (2017). Variability of temperature and its impact on reference evapotranspiration: The test case of the Apulia Region (Southern Italy). Sustainability, 9(12), 2337.

·         Giorgio G. A., Ragosta M., Telesca V. (2016). Application of a multivariate statistical index on series of weather measurements at local scale. 112, 61-66. Measurement, https://doi.org/10.1016/j.measurement.2017.08.005

·         Gan, Y., Liang, X. Z., Duan, Q., Ye, A., Di, Z., Hong, Y., & Li, J. (2018). A systematic assessment and reduction of parametric uncertainties for a distributed hydrological model. Journal of hydrology, 564, 697-711.

·         Coluzzi, R., D’Emilio, M., Imbrenda, V., Giorgio, G. A., Lanfredi, M., Macchiato, M., ... & Telesca, V. (2019). Investigating climate variability and long-term vegetation activity across heterogeneous Basilicata agroecosystems. Geomatics, Natural Hazards and Risk, 10(1), 168-180.

ü  Line 90-91 "The meteorological data includes temperature data from 12 weather stations and precipitation data from 16 rainfall stations (12 meteorological stations and 4 hydrological stations). Please explicit the timing of meteorological data used.

ü  Line 98 "hydroinformatic modeling system". Please enter uppercase letters.

ü  The description of "Analysis on climate change" in methodology section requires some little improvements. Explain the main steps followed and possible feedback in the proposed methodological approach.

Author Response

Dear Review:

We appreciated very much to Reviewers for your positive comments, the reviewers’ comments and suggestions are very important to improve the manuscript, and the authors thank the reviewers a lot. The relative corrections were listed below.

1. In the introduction section, the authors are encouraged to revisit the literature review and to expand on some recent key studies around the "climate change" thematic, improving the background. Moreover I suggest you to mention the importance of studying the impact of climate change to local scale. The following studies could be a good source of references:

It has been modified in lines 34-54 in the revised manuscript.

2. Line 90-91 "The meteorological data includes temperature data from 12 weather stations and precipitation data from 16 rainfall stations (12 meteorological stations and 4 hydrological stations). Please explicit the timing of meteorological data used.

It has been added in lines 94-95 in the revised manuscript.

3. Line 98 "hydroinformatic modeling system". Please enter uppercase letters.

It has been modified in line 101 in the revised manuscript.

4. The description of "Analysis on climate change" in methodology section requires some little improvements. Explain the main steps followed and possible feedback in the proposed methodological approach.

It has been added in lines 197-224 in the revised manuscript.

Round 2

Reviewer 1 Report

See attached file.

Author Response

Dear Reviewer:

We appreciated very much to Reviewers for your positive comments, the reviewers’ comments and suggestions are very important to improve the manuscript, and the authors thank the reviewers a lot. The relative corrections were listed below.

1. The authors have improved the manuscript according to the reviewer’s comments. Only one issue remains open. In my opinion Eq, 12 has to be rewritten in this way:

It has been modified in line 186 in the revised manuscript. 
